# Lower Percentage of Fat Mass among Tai Chi Chuan Practitioners

**DOI:** 10.3390/ijerph17041232

**Published:** 2020-02-14

**Authors:** Silvia Stagi, Azzurra Doneddu, Gabriele Mulliri, Giovanna Ghiani, Valeria Succa, Antonio Crisafulli, Elisabetta Marini

**Affiliations:** 1Department of Life and Environmental Sciences, University of Cagliari, Cittadella Universitaria, Monserrato, 09042 Cagliari, Italy; valerias@unica.it; 2Department of Medical Sciences and Public Health, University of Cagliari, 09124 Cagliari, Italy; didi-zazzy@hotmail.it (A.D.); jabutele84@gmail.com (G.M.); giovanna.ghiani@tiscali.it (G.G.); crisaful@unica.it (A.C.)

**Keywords:** ageing, Tai Chi Chuan, *specific* bioelectrical impedance vector analysis (BIVA), body composition

## Abstract

The aim of the study was to analyze total and regional body composition in Tai Chi Chuan (TCC) middle-aged and elderly practitioners. A cross-sectional study on 139 Italian subjects was realized: 34 TCC practitioners (14 men, 20 women; 62.8 ± 7.4 years) and 105 sedentary volunteers (49 men, 56 women; 62.8 ± 6.4 years). Anthropometric measurements (height, weight, arm, waist, and calf circumferences), hand-grip strength, and physical capacity values were collected. Total and regional (arm, leg, and trunk) body composition was analyzed by means of *specific* bioelectrical impedance vector analysis (*specific* BIVA). TCC practitioners of both sexes were characterized by a normal nutritional status, normal levels of physical capacity, and normal values of hand-grip strength. Compared to controls, they showed lower percentages of fat mass (lower *specific* resistance) in the total body, the arm, and the trunk, and higher muscle mass (higher phase angle) in the trunk, but lower muscle mass in the arm. Sexual dimorphism was characterized by higher muscle mass (total body, arm, and trunk) and lower %FM (arm) in men; sex differences were less accentuated among TCC practitioners than in the control. TCC middle-aged and elderly practitioners appear to be less affected by the process of physiological aging and the associated fat mass changes, compared to sedentary people.

## 1. Introduction

Aging is associated with body composition variations: muscle mass progressively declines, while fat mass (FM) initially increases, especially in the visceral region, and then levels off or decreases [1]. These variations may lead to sarcopenia, a condition characterized by both low skeletal muscle mass and low skeletal muscle strength or quality, and to sarcopenic obesity, due to the coexistence of sarcopenia and fat excess or fat infiltration into muscle [2]. Sarcopenic and sarcopenic obese elderly have a lower independence and an increased risk of morbidity and mortality [2].

Physical activity has been shown to contribute substantially to the maintenance of the individual physiological and psychological well-being in all phases of life [3]. In the elderly, physical exercise can slow down or even reverse the trend towards sarcopenia [4,5]. However, which kind of physical activity could provide the more beneficial effects is still unclear and needs to be studied in depth [3]. Furthermore, body composition changes in different body districts were still poorly studied in the elderly. Instead, regional body composition gives relevant information. In particular, it allows better understanding of the role of physical activity in different conditions, such as sarcopenia and obesity [6].

Tai Chi Chuan (TCC) is an ancient Chinese martial art. The practice consists of the repetition of a sequence of slow and harmonious movements, focused on balance and based on respiration techniques. This discipline is particularly adequate for elderly people, who find difficult to perform rapid movements, and has been recommended to improve quality of life and to prevent or retard sarcopenic obesity [4]. TCC practice may help improve coordination and balance [7], retard bone loss [8], maintain cardiorespiratory fitness and flexibility [9]. It also contributes to promote social and psychological health [10], and cognitive function [11].

The effect of TCC on total and regional body composition has been poorly studied and the literature shows a still unclear pattern. Dual-energy X-ray absorptiometry (DXA) has been used to assess bone mass density in postmenopausal TCC women [8], whereas anthropometric techniques [9,12], equations based on bioimpedance measures [12,13,14,15,16], or air displacement [17] have been applied to analyze fat and fat-free mass. The studies on regional body composition have been based on skinfold thickness distribution only [12]. *Specific* bioelectrical impedance vector analysis (*specific* BIVA; [18]) has never been used. *Specific* BIVA is based on the analysis of bioelectrical values (resistance, R; reactance, Xc; Ω), standardized by body height and transversal cross-sections, in order to minimize the effect of conductor dimensions, that is the effect of anthropometric differences. Bioelectrical vectors can be projected on the Cartesian plane, where they are defined by their length (impedance: (R^2^ + Xc^2^)^0.5^) and inclination angle (phase angle: arctan Xc/ R180/π). *Specific* BIVA has been validated against DXA in a sample of US adults [19], showing high sensitivity and specificity in the evaluation of %FM (the longer the vector, the higher the %FM). It has also shown to be highly correlated with DXA results in elderly subjects [20,21] and young athletes [22]. Furthermore, phase angle, a variable considered a proxy of muscle mass [23], has shown to be positively correlated with intracellular/extracellular water ratio (ICW/ECW), when compared to dilution techniques [22,24]. Indeed, *specific* BIVA has been proposed as a promising technique for body composition assessment in athletes [25], and has been already applied in different contexts (e.g., cavers: [26]; various athletes: [22]; soccer players: [27]).

The aim of the present research was to analyze total body and regional body composition in TCC middle-aged and elderly practitioners by means of *specific* bioelectrical impedance vector analysis.

## 2. Materials and Methods

This observational study was realized on a cross-sectional sample of TCC practitioners and an age-matched group of sedentary subjects. The measurement process flow chart is shown in Figure 1.

### 2.1. Subjects

The sample was composed of 34 middle-aged and elderly TCC volunteers (14 men and 20 women) aged 62.8 ± 7.4 years, recruited from the A.S.D. Tai Chi Chuan school of Cagliari and La Porta d’Oriente of Quartu S.Elena (Italy). At the time of the measurement, the subjects had already been practicing Tai Chi Chuan for an average of six years and were training three times a week or more. Four of them participated in other sports too (swimming, walking, yoga, cycling). All the subjects declared to eat a balanced diet, with a great intake of fruits and vegetables.

The control group was composed of 105 volunteers (49 men, 56 women) of the same mean age (62.8 ± 6.4 years), and living in the same geographical area, selected for not practicing physical exercise.

Criteria of exclusion were physical handicaps, pathologies that might influence the measurements, metallic prostheses, pacemakers, or limb amputations.

Data was deposited in the University of Cagliari repository: http://hdl.handle.net/11584/269226.

The research was approved by the Independent Ethical Committee of the A.O.U. of Cagliari (PG/2017/1700). Each participant was informed about the purposes and methods of the research and signed consent to participate.

### 2.2. Measurements

#### 2.2.1. Anthropometry

Anthropometric measurements (height, cm; weight, kg; waist circumference, cm; arm and calf circumferences of both sides) were taken using international standard procedures [28]. The length (cm) of each regional district was also recorded: arm length, defined as the acromion-stylion distance; leg length, as the distance between the great trochanter and the malleolus; trunk length, as the distance between injector electrodes.

Body mass index BMI was calculated by the formula: weight/height^2^ (kg/m^2^). Underweight, normal weight, overweight and obesity, and visceral obesity were defined according to BMI [29].

#### 2.2.2. Bioimpedance

Whole body and regional bioelectrical impedance measurements (resistance, R, and reactance, Xc, at 50 kHz and 800 μA) were taken using a single-frequency phase sensitive impedance analyzer (BIA 101, Akern, Firenze, Italy).

Following the European Society for Clinical Nutrition and Metabolism Working Group guidelines [30], the measurements were taken in the morning and the volunteers were asked to avoid drinking and eating (3 h before the test; alcohol 24 h), exercising (12 h before the test), and to void their bladder before the examination. The device was checked before each session with a calibrated circuit whose impedance values are known: R = 380 Ω, Xc = 47 Ω (±2% error). The intra-observer technical error of measurement (TEM) and the %TEM were calculated in a subsample of 25 subjects (R: TEM = 3.5 ohm, TEM% = 0.6%), that are within the admitted variability of the device. The accuracy of *specific* BIVA in evaluating %FM has been showed by Buffa et al. [19] using DXA as a reference (receiving operator curves, ROC areas: 0.84–0.92), while Marini et al. [22] showed the agreement with dilution techniques in the evaluation of ICW/ECW.

Bioimpedance measurements were taken with the subject lying supine, on the right side of the total body and the trunk, the right arm and leg, using two pairs of detector and injector electrodes. For total body measurements, on the hand, the injector was placed at the distal extremity of the third metacarpal, and the detector on the dorsal surface of the wrist, at level of the styloid process; on the foot, the injector was placed at the base of the second and third metatarsals, and the detector on the dorsal surface, at the median point of the tibial tarsal joint. Regional bioelectrical measurements were taken using the following procedure. Arm: on the shoulder, the injector was placed at the acromion process level and the detector at 5 cm distance, following the axillary line [31]; on the hand, the same position of the total body was used. Leg: on the hip, the injector was placed anterior to the iliac crest and the detector at a distance of 5 cm [32]; on the foot, the same position of the total body was used. Trunk: the electrodes already positioned on the shoulder and hip for the regional measures were used.

*Specific* bioelectrical impedance vector analysis [18] was used to estimate body composition. Resistance and reactance were multiplied by a correction factor (A/L). For the total body, the A value was estimated as 0.45 arm area + 0.10 trunk area + 0.45 calf area (cm^2^); arm, trunk, and calf area were calculated as C^2^4π, where C (cm) is the circumference of the segment. The length was calculated as L = 1.1H, where H is the height in cm. In the regional approach, A/L for the arm, leg, and trunk were calculated considering the cross section of the mid arm, calf, and wrist (A) and the arm, leg, and trunk length (L), respectively.

*Specific* impedance (Zsp) was calculated as (Rsp^2^ +Xcsp^2^)^0.5^ (Ω cm) and phase angle as arctan Xc/R180/π (degree).

#### 2.2.3. Hand-Grip Strength

Hand-grip strength was taken using a Sahean hand dynamometer (Hydraulic Hand Dynamometer Saehan Corporation, MSD buba Belgium). The volunteers were asked to hold the dynamometer with the elbow flexed at 90° and to squeeze it with maximum isometric effort for a few seconds, three times with a 1 min interval between each attempt. Only the highest strength value was taken into account. According to Dodds et al. [33], 27 kg for men and 16 kg for women were considered the diagnostic threshold for assessing probable sarcopenia.

#### 2.2.4. Mini-Nutritional Assessment

The Mini Nutritional Assessment (MNA^®^; [34]) is a recommended multidimensional method to evaluate nutritional status in the elderly, and was applied to the subgroup of subjects over 60 years old only. It is an 18 item questionnaire considering: dietary habits, living conditions, anthropometry, cognitive and disability status. A normal nutrition score is 24 or higher, a score between 17 and 23.5 designates risk of malnutrition, and a score lower than 17 indicates a condition of malnutrition.

#### 2.2.5. Physical Capacity Assessment

Participants underwent a cardiopulmonary test (CPX) with a gas analyzer (ULTIMA CPX, MedGraphics St. Paul, MN, USA), while pedaling on a mechanically braked cycle ergometer (CUSTO Med, Ottobrunn, Germany). The test consisted of a linear increase of workload (10 W/min), starting at 20 W, at a pedaling frequency of 60 rpm, until exhaustion, which was considered the point when the subject was unable to maintain a pedaling rate of at least 50 rpm. During the CPX, the following variables were gathered: oxygen uptake (V̇O_2_), carbon dioxide production (V̇CO_2_), respiratory exchange ratio (RER, calculated as V̇CO_2_/V̇O_2_), pulmonary ventilation (VE), and heart rate (HR). The oxygen pulse (V̇O_2_/HR), a parameter related to stroke volume and cardiac performance [35], was also calculated. Workloads reached at anaerobic threshold (WAT) and at maximum (Wmax) were taken into consideration. AT was calculated using the V-slope method, while V̇O_2_ at Wmax (V̇O_2_max) was calculated as the average V̇O_2_ during the final 30 s of the incremental test. Achievement of V̇O_2max_ was considered as the attainment of at least two of the following criteria: (1) a plateau in V̇O_2_ despite increasing workload (<80 mL·min^−1^); (2) RER above 1.10; and (3) HR ± 10 beats·min^−1^ of predicted maximum HR calculated as 220-age [36].

### 2.3. Statistical Analysis

The comparison between anthropometric and bioelectrical measurements in the TCC group and controls, considering sex, was realized using a two-way ANOVA. Cohen’s d [37] for independent samples was calculated to measure the effect size in the comparison between TCC practitioners and controls.

Confidence ellipses, representing the area around the sample mean within which the “true mean” is expected to lie with a probability of 95%, and Hotelling’s T^2^ test were used to compare total body composition characteristics in the TCC and control samples, and between sexes.

The SPSS program was used to calculate univariate and bivariate statistics. *Specific* BIVA was applied using an ad hoc software available online (www.specificbiva.unica.it).

## 3. Results

Men and women practicing Tai Chi Chuan showed in mean a normal weight condition, according to BMI, and were below the threshold for visceral adiposity, according to waist circumference (Table 1). A good nutritional status was observed also in the individuals aged more than 60 years (MNA: men, 27.3 ± 1.0; women, 27.5 ± 0.6); no subjects were at risk of malnutrition. Compared to the age-matched control sample, TCC men and women showed higher height, lower circumferences, and lower BMI. They also showed lower total body and arm *specific* resistance, higher phase angle in the trunk, and lower phase angle in the arm (Table 1, Figure 2 and Figure 3). According to *specific* BIVA, these values indicate lower %FM in the total body and the arm, and higher ICW/ECW and muscle mass in the trunk, but not in the arm. Furthermore, among the controls there was a high prevalence of individuals at risk of malnutrition (18%). Cohen’s d showed large or medium size effects on anthropometric (with the exception of stature) and on total body and arm bioelectrical comparisons (with the exception of total body phase angles), while the effect was small on leg and trunk bioelectrical measurements, hand-grip strength, and age.

Hand-grip strength (HGS) values were above the cut offs for sarcopenia in both TCC and controls.

Table 2 reports values of cardiopulmonary parameters gathered during the CPX test.

Both TCC and control men showed higher values than women in all anthropometric measurements, except for BMI, higher values of hand-grip strength and of total body, arm and trunk phase angles, and lower values of arm *specific* resistance and impedance (Table 1). Sex differences in bioelectrical values were less accentuated among TCC practitioners than in the controls. In fact, bioimpedance mean vectors were significantly different between in the control group only (Figure 4).

## 4. Discussions

The present research showed that TCC practitioners of both sexes, including the older ones, were overall characterized by a normal nutritional status, normal levels of physical capacity, and normal values of hand-grip strength. With respect to a sample of sedentary people of similar age, they showed lower percentages of fat in the total body, the arm, and the trunk (as indicated by the lower specific vector length and waist circumferences), and higher muscle mass in the trunk (as indicated by the higher phase angle), but lower muscle mass in the arm, and similar hand-grip strength. Sexual dimorphism was characterized by higher muscle mass (total body, arm, and trunk) and hand-grip strength, and lower %FM (arm) in men; sex differences were less accentuated among TCC practitioners than in the control.

These results suggest a positive effect of TCC practice on body composition, particularly on fat mass. Consistently with our results, the literature focused on body composition in TCC suggest a major effect of TCC on fat mass. Lan et al. [9] observed a lower percentage of body fat among TCC practitioners. Longitudinal studies on the effect of a 10/12-week Tai Chi program showed a reduction of %FM [13,15], or of body fat [14,17]. The analysis of regional body composition and anthropometry confirmed the effect of TCC on %FM. The lower %FM values (lower specific vector length) among TCC practitioners with respect to controls mainly concerned the arm and the trunk, where a lower waist circumference, indicative of lower visceral fat accumulation, was also detected. These results are indicative of a healthy body composition, considering the role of visceral fat in the development of metabolic disorders and cardiovascular diseases [38]. It should also be noted that such phenotype is unexpected, considering the increase of abdominal adiposity commonly associated with physiological aging [39]. The body composition of the legs, instead, was not significantly different from the control. Conversely, Yu et al. [12] detected lower values of subcutaneous adipose tissue in the thighs of TCC practitioners than in the control.

Overall, our results also suggest a weak effect of TCC on muscle mass, as indicated by the similar hand-grip strength and similar or even lower phase angle of TCC practitioners with respect to the control group. Similarly, Yu et al. [12] and Lai et al. [16] failed to detect differences in the comparison with a sample of swimmers, nor respect to the control [12], and Kelly and Gilman [4] observed that TCC was not as effective at building muscle compared to other types of physical exercise. However, other studies showed an improvement of skeletal muscle mass [13,14], of muscle strength [13], and of functional performance [40,41], and suggested that TCC would have a similar effect as walking on parameters related to aerobic metabolism [9,15].

The regional approach of this study also allowed a detailed analysis of sex differences. The observed larger muscle mass in men and the higher %FM in the women, particularly in the arm and leg, are coherent with the known pattern of sexual dimorphism [6]. Furthermore, the higher waist circumference values of men are consistent with their generally greater central fat distribution. However, mean waist values were below the threshold for visceral obesity in both sexes, consistently with the results on low visceral adiposity reported among TCC subjects in this study and by other authors [12]. Furthermore, sex differences were less accentuated among TCC practitioners than in the control. This finding is not consistent with the low degree of sexual dimorphism detected in young subjects practicing different kinds of physical activities [42].

On the whole, the results of this study suggest that TCC practitioners do not strictly follow the trend towards the increase of fat mass, particularly of visceral adiposity, the reduction of hand-grip strength, and the worsening of nutritional status generally observed among middle-aged and elderly people [1]. Hence, TCC practitioners appear to be less affected by the process of physiological aging and the associated fat mass and functional changes. The healthy lifestyle commonly adopted by TCC practitioners could also contribute to such pattern.

This study has points of strength. In fact, this is the first study on total body and regional body composition in middle-aged and elderly athletes of both sexes using, phase angle, and *specific* BIVA, that is a procedure particularly promising for body composition analysis in sport [25]. The main limitation of the study is related to the sample size of TCC practitioners, which is not so large due to the peculiarity of the age range and the still limited diffusion of the discipline in Western countries. However, the information on regional body composition, that is scant in middle-aged and elderly athletes, can be insightful for future research.

## 5. Conclusions

The present research showed that TCC practitioners were characterized by good nutritional status (normal values of BMI, MNA, and low levels of fat mass in the total body, in the arm, and in the trunk) and had normal values of hand-grip strength. The effect on muscle mass was less evident. These results confirm the positive effect of TCC practice on body composition and functionality. Considering the suitableness of the practice in late adulthood over other disciplines, TCC represents a safe and effective way to help improve nutritional status and physical function.

From a methodological point of view, *specific* BIVA appears a suitable technique to screen and monitor total body and regional body composition in middle-aged and elderly subjects, in order to evaluate their nutritional status and risk of morbidity.

## Figures and Tables

**Figure 1 ijerph-17-01232-f001:**
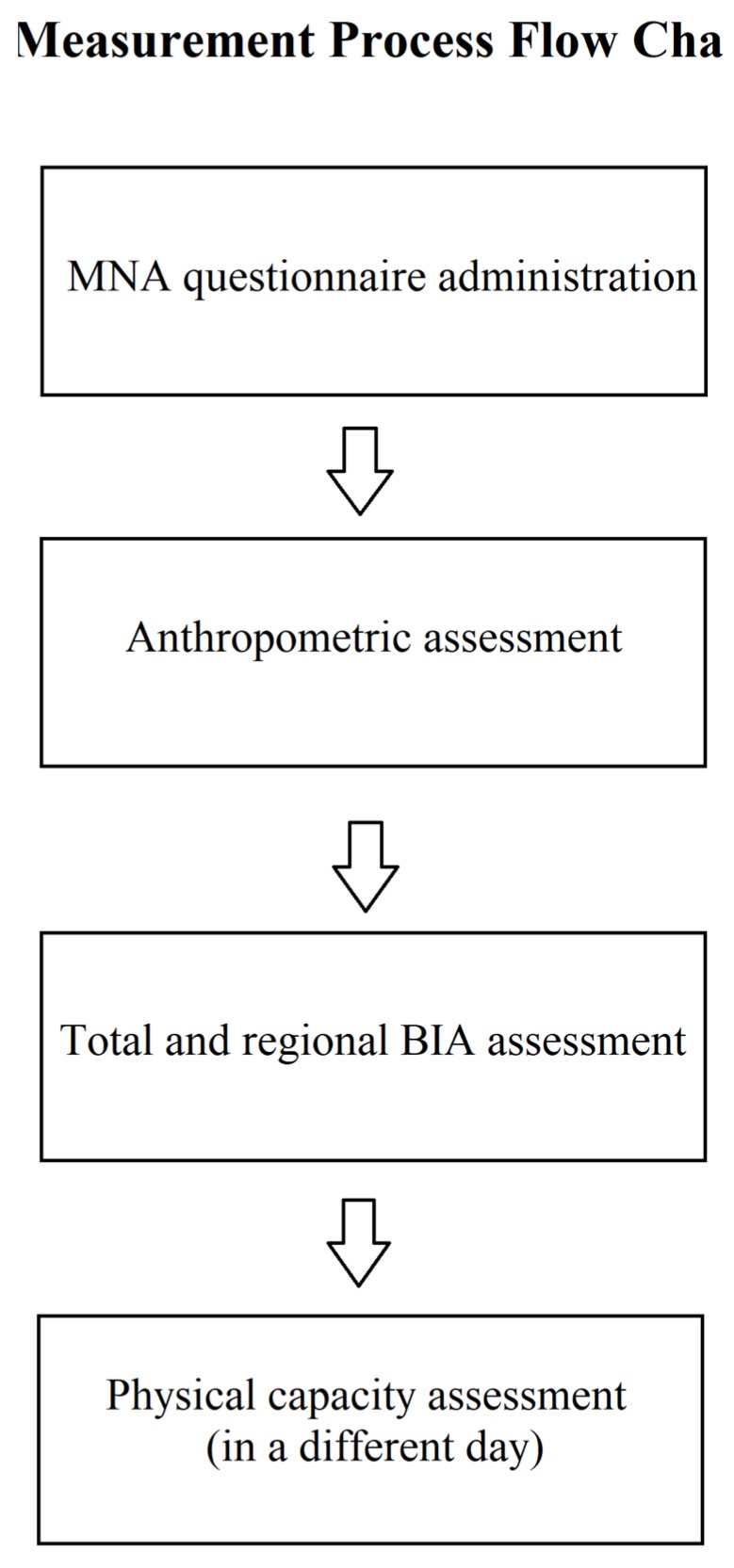
Measurement process flow chart.

**Figure 2 ijerph-17-01232-f002:**
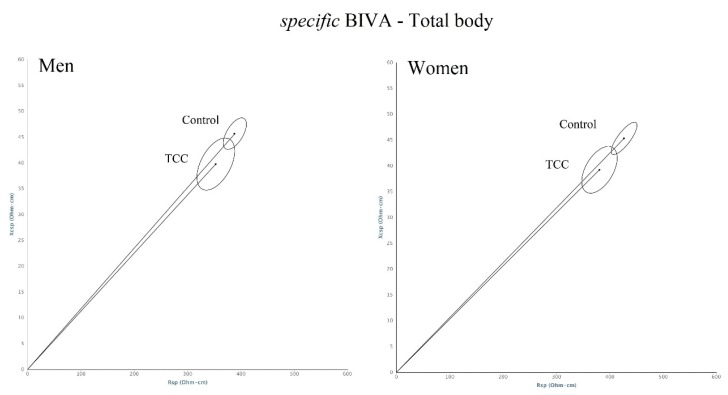
Total body confidence ellipses. Comparison between the Tai Chi Chuan (TCC) group and controls. Men: T^2^ = 6.6, *p* = 0.047; women: T^2^ = 8, *p* = 0.023.

**Figure 3 ijerph-17-01232-f003:**
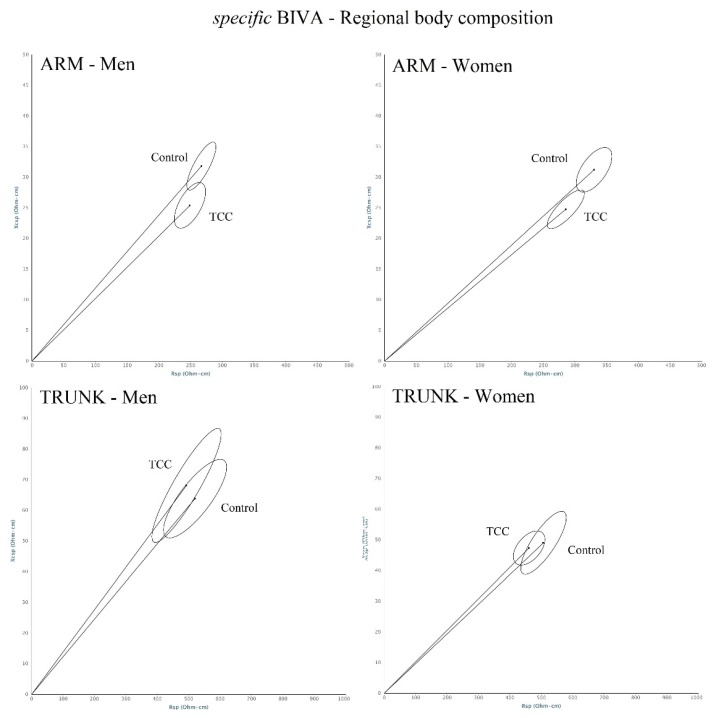
Regional confidence ellipses. Comparison between TCC group and controls. Arm (men: T^2^ = 9.8, *p* = 0.001; women: T^2^ = 11.2, *p* = 0.007); trunk (men T^2^ = 2.7, *p* = 0.285; women T^2^ = 2.2, *p* = 0.340).

**Figure 4 ijerph-17-01232-f004:**
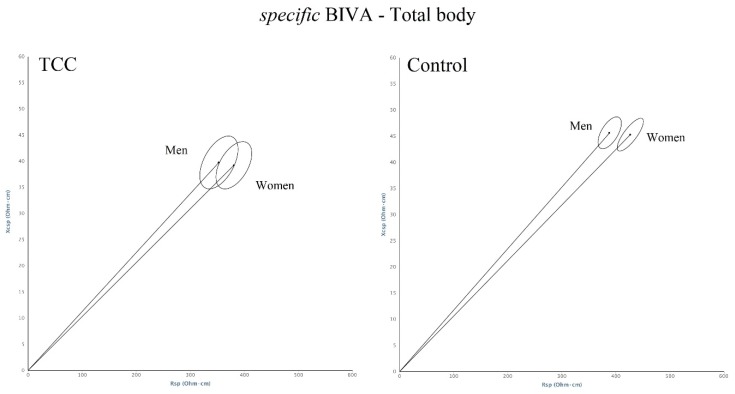
Total body confidence ellipses. Comparison between sexes. TCC: T^2^ = 3, 8, *p* = 0.177; controls: T^2^ = 20.1, *p* = 0.000.

**Table 1 ijerph-17-01232-t001:** Anthropometry, total and regional body composition in the Tai Chi Chuan group. Two-way ANOVA for the comparison between sexes and with the control sample of healthy Italian adults.

Total Body Composition	Tai Chi Chuan				Control				F		
	Men (*n* = 14)		Women (*n* = 20)		Men (*n* = 49)		Women (*n* = 56)				
	Mean	s.d.	Mean	s.d.	Mean	s.d.	Mean	s.d.	*p* _sex_	*p* _sport_	*p* _sex•sport_
**Age (y)**	63.4	7.9	62.5	7.1	62.9	6.6	62.8	6.4	0.713	0.948	0.763
**Height (cm)**	172.2	5.1	155.5	6.7	166.1	6.2	153.2	6.7	0.000	0.001	0.151
**Weight (kg)**	70.1	7.2	54.1	7.4	77.8	11.8	64.0	11.5	0.000	0.000	0.615
**BMI (kg/m^2^)**	23.7	2.5	22.4	3.0	28.2	4.1	27.3	4.7	0.186	0.000	0.824
**Waist (cm)**	87.2	9.0	74.0	6.9	97.4	11.3	85.2	12.1	0.000	0.000	0.822
**Arm (cm)**	28.6	2.1	26.0	2.2	30.3	3.1	29.0	3.2	0.000	0.000	0.297
**Calf (cm)**	35.2	2.0	33.0	2.4	36.8	3.3	34.9	3.7	0.002	0.009	0.876
**Rsp (ohm·cm)**	352.7	45.6	380.4	54.2	454.9	56.3	553.2	50.5	0.010	0.001	0.677
**Xcsp (ohm·cm)**	39.7	6.5	39.2	7.4	53.4	8.9	58.8	9.1	0.794	0.001	0.955
**Zsp (ohm·cm)**	355.0	45.6	382.5	54.3	391.12	59.4	429.0	71.8	0.011	0.001	0.680
**Phase angle sp (°)**	6.5	1.0	5.9	1.0	6.7	1.0	6.1	0.8	0.001	0.275	0.756
**Regional Body Composition**	**Men (*n* = 14)**		**Women (*n* = 20)**		**Men (*n* = 27)**		**Women (*n* = 33)**		***p*_sex_**	***p*_sport_**	***p*_sex•sport_**
**Arm: R sp (ohm·cm)**	248.8	31.7	285.4	47.9	266.9	44.7	330.0	61.9	0.000	0.005	0.230
**Arm: Xc sp (ohm·cm)**	25.4	4.8	24.7	5.2	31.8	7.65	31.2	8.0	0.687	0.000	0.993
**Arm: Z sp sp (ohm·cm)**	250.1	31.8	286.4	48.1	268.8	45.03	331.6	62.0	0.000	0.005	0.235
**Arm: phase angle sp (°)**	5.8	1.0	5.0	0.6	6.8	1.1	5.4	1.1	0.000	0.002	0.287
**Leg: R sp (ohm·cm)**	254.6	17.4	287.0	34.2	255.2	44.4	297.9	58.52	0.000	0.562	0.601
**Leg: Xc sp (ohm·cm)**	27.3	3.6	29.6	8.0	31.3	9.6	31.7	9.9	0.476	0.116	0.621
**Leg: Z sp sp (ohm·cm)**	256.1	17.4	288.7	33.7	257.2	45.0	299.7	58.9	0.000	0.545	0.619
**Leg: phase angle sp (°)**	6.1	0.8	6.0	2.2	6.9	1.4	6.03	1.3	0.135	0.229	0.241
**Trunk: R sp (ohm·cm)**	492.9	142.5	436.5	132.4	520.0	196.8	506.7	159.3	0.523	0.305	0.791
**Trunk: Xc sp (ohm·cm)**	68.1	24.1	44.8	14.0	63.8	25.1	49.0	22.6	0.000	0.793	0.538
**Trunk: Z sp sp (ohm·cm)**	497.7	144.1	438.9	132.9	524.0	198.0	509.2	160.4	0.491	0.313	0.783
**Trunk: phase angle sp (°)**	7.8	1.4	5.9	1.0	7.1	1.4	5.4	1.4	0.000	0.048	0.700
**Hand grip (kg)**	39.6	8.9	24.3	5.0	38.0	9.4	23.6	5.4	0.000	0.789	0.913

s.d.: standard deviation; BMI: body mass index; Rsp: *specific* resistance; Xcsp: *specific* reactance; Zsp: *specific* impedance.

**Table 2 ijerph-17-01232-t002:** Results of the cardiopulmonary test (CPX) test for the whole group (top table), and for women (middle table) and men subgroups (bottom panel).

**Average Result for the Whole Group**								
	**Workload (w)**	**V̇O_2_ (mL/kg·min^−1^)**	**V̇O_2_ (mL·min^−1^)**	**V̇CO_2_ (mL·min^−1^)**	**RER**	**VE (L·min^−1^)**	**HR (bpm)**	**OP (mL·bpm^−1^)**
**Rest**	0	4.05 ± 0.79	257.5 ± 57.5	249.1 ± 73.5	0.96 ± 0.16	9.1 ± 2.8	77.6 ± 13.7	3.4 ± 1.1
**WAT**	90.7 ± 40.0	18.03 ± 5.41	1153.4 ± 409.8	1303.6 ± 522.6	1.12 ± 0.13	35.8 ± 12.4	130.9 ± 24.2	8.8 ± 2.5
**Wmax**	119.4 ± 49.4	22.76 ± 6.63	1449.3 ± 473.3	1824.3 ± 669.5	1.25 ± 0.13	54.9 ± 18.9	150.3 ± 25.6	9.7 ± 2.8
**Average Results for Women Subgroup**								
**Rest**	0	4.08 ± 0.90	233.5 ± 52.7	228.9 ± 86.7	0.96 ± 0.21	8.2 ± 2.9	80.0 ± 12.5	2.9 ± 0.8
**WAT**	73.1 ± 30.8	18.91 ± 4.09	976.12 ± 267.3	1075.1 ± 286.9	1.11 ± 0.16	29.4 ± 7.9	131.1 ± 19.7	7.4 ± 1.5
**Wmax**	92.7 ± 32.5	19.75 ± 3.90	1142.8 ± 278.6	1429.3 ± 361.3	1.25 ± 0.42	43.0 ± 12.4	148.3 ± 18.2	7.7 ± 1.6
**Average Results for Men Subgroup**								
**Rest**	0	4.14 ± 0.84	291.0 ± 52.4	287.4 ± 47.2	0.99 ± 0.11	10.7 ± 2.1	73.5 ± 15.0	4.1 ± 1.1
**WAT**	95.4 ± 33.7	16.99 ± 4.79	1194.2 ± 311.5	1375.9 ± 435.4	1.14 ± 0.11	39.3 ± 11.8	122.0 ± 27.1	9.8 ± 2.1
**Wmax**	128.6 ± 44.8	22.81 ± 6.09	1583.6 ± 309.0	1995.4 ± 566.9	1.24 ± 0.15	61.5 ± 18.7	140.7 ± 28.0	11.4 ± 1.7

WAT = workload at anaerobic threshold; Wmax = maximum workload; V̇O_2_ = oxygen uptake indexed by body weight; V̇O_2_ = oxygen uptake; V̇CO_2_ = carbon dioxide production; RER = respiratory exchange ratio; VE = pulmonary ventilation; HR = heart rate; OP = oxygen pulse.

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
