# Peer review of "Lower Percentage of Fat Mass among Tai Chi Chuan Practitioners"

_ijerph, 2020, doi:10.3390/ijerph17041232_

Round 1

Reviewer 1 Report

The main objective of the study entitled Total and Regional Body Composition among Tai Chi Chuan practitioners is to analyze the total and regional body composition in middle-aged and older CBT practitioners through a specific analysis of the bioelectric impedance vector.

The introduction presents an adequate structure and a good justification of the reasons for the study.

However, the method has a number of shortcomings, for example, the procedure followed in the study is not completely clear to me, so I suggest the authors to be more explicit.

In relation to the participants of the study, I lack information, above all that related to their habits, if they practiced any type of previous physical activity, if they practiced any sport, eating habits, etc. All these variables can have a great influence on the final results of the study.

Regarding the analyses there are several things that concern me.

The main one is that the authors must present the analyses to know if the experimental and control groups start with a difference before the beginning of the study. If this analysis has not been taken into account, please justify why.

As for the ANOVA analyses, I miss the Cohen's d in order to know the effect size of the results and thus to know if the changes have been low, moderate or large (Coe and Merino, 2003; Cohen, 1988).

Coe, R. y Merino, C. (2003). Effect Size: A guide for researchers and users. Revista de Psicología, 21(1), 147-177.

Cohen, J. (1988). Statistical Power Analysis for the Behavioral Sciences. (2nd ed.), New Jersey: Lawrence Erlbaum Associates.

Author Response

Thank you for the comments and useful suggestions.

> The method has a number of shortcomings, for example, the procedure followed in the study is not completely clear to me, so I suggest the authors to be more explicit.

- Following this suggestion, we have described more clearly the nature of the study (Methods: lines 79-80), also adding a new figure (new figure 1) with the measurements process flow chart, as suggested by Reviewer#3.

> In relation to the participants of the study, I lack information, above all that related to their habits, if they practiced any type of previous physical activity, if they practiced any sport, eating habits, etc. All these variables can have a great influence on the final results of the study.

- As suggested, we have added information on TCC subjects and controls (Methods: lines 86-88; 90-91), and a comment on the possible influence of dietary habits (Discussion: lines 54-55).

> Regarding the analyses there are several things that concern me. The main one is that the authors must present the analyses to know if the experimental and control groups start with a difference before the beginning of the study. If this analysis has not been taken into account, please justify why.

- We are sorry, but actually we do not understand this comment. This is not a longitudinal study and we have not a start point before the beginning of the study. In our cross-sectional sample, we can only observe the differences between individuals practicing TCC and the control group of sedentary people. We hope that the more detailed procedure description of the new version will help to solve the Reviewer's concerns.

>As for the ANOVA analyses, I miss the Cohen's d in order to know the effect size of the results and thus to know if the changes have been low, moderate or large.

- Thank you for this suggestion. We have added the Cohen's d, that allowed us to confirm the results of ANOVA statistics (Methods: lines 181-183; Results: lines 196-199).

Reviewer 2 Report

This is an interesting and meaningful study. It was concluded that middle-aged and elderly people participating in Tai Chi Chuan (TTC) can have good BMI, NMA and normal grip strength. In terms of research and expression of the manuscript, the following suggestions are provided for the authors.

Comments

This study extensively discussed the effects of TTC on body composition, such as total body fat rate, skeletal muscle mass, subcutaneous fat, visceral fat, etc. However, the body composition items mentioned above were not actually measured in this study. So a considerable amount of correction is needed in your discussion section.

For the above reasons, the impact of TCC on body composition was mentioned in the title, abstract, and conclusion of this study.
From the results, we can only see the effect of TCC on Anthropometric measurements(BMI, waist, arm, calf). It is suggested that the title, abstract, discussion and conclusions of this study need to be significantly revised.

In this study, assessments and measurements of BIVA, Hand-grip strength, Min-nutritional assessment, and physical capacity assessment have been performed. However, the results were only presented in the table, and the relationship between the TCC and the control group has not been analyzed and discussed. It is suggested that the discussion, title, and conclusion of this research can be carried out in this regard. It was not just about the relationship between TCC and body composition. Mainly because the body composition measurement was not actually performed in this study.

In Table 1, the phase angle TTC of the female was 5.9 degrees, the control group was 6.1 degrees, the Arm phase angle of the male TTC was 5.8 degrees, the control group was 6.8 degrees, the male TTC leg phase angle was 6.1 degrees, and the control group was 6.9 degrees. Some TTCs in the table have better body angle than the control group, and some did not. Most studies have shown that the healthier the body, the higher the cell viability, but the results here need further discussion and clarification.

Please provide the IRB number.

In the BIA test conditions, although it was stated that the experiment was performed in the morning, drinking and diet control, and urination control.
When conducting BIVA experiments, there are still many restrictions that need to be clarified, and the author is recommended to enhance the related narrative.

In the manuscript, the author only mentions the participation status at TTC, and did not mention the exclusion conditions or disease status of the test subjects, and it is recommended to be added.

Please enhance the drawing quality or resolution.

For the measurement of the bioimpedance parameters of each limb segment, it must be clearly described, and its reliability and repeatability should be explained.
What was the basis for determining the sample size in this study?

Author Response

Thank you for the comments and useful suggestions.

> The body composition items mentioned above were not actually measured in this study. So a considerable amount of correction is needed in your discussion section. [...] we can only see the effect of TCC on anthropometric measurement, waist, arm, calf). It is suggested that the title, abstract, discussion and conclusions of this study need to be significantly revised.

- We respectfully disagree with this comment, possibly due to a lack of clarity in the first version of our manuscript. Specific BIVA is a recognised method for the evaluation of body composition. It has been validated against DXA in a sample of adults (Buffa et al., 2013), showing high sensitivity and specificity in the evaluation of FM% (the longer the vector, the higher the %FM). It has also shown to be highly correlated with DXA results in samples of elderly subjects (Marini et al., 2013; Saragat et al., 2014) and in young athletes (Marini et al., 2019). Furthermore, phase angle, a variable considered a proxy of muscle mass (Norman et al., 2012), has shown to be positively correlated with intracellular/extracellular water ratio (ICW/ECW), when compared to dilution techniques (Marini et al., 2019; Campa et al., 2019). For these reasons, a recent review has recommended specific BIVA as a particularly useful technique for the analysis of body composition in athletes (Castizo-Olier et al., 2018). The technique has been already applied in different studies (e.g., cavers: Antoni et al., 2017; various athletes: Marini et al., 2019; soccer players: Campa et al., 2020).

In the revised version of the manuscript, we have added a more detailed and hopely clearer explanation of the principles of specific BIVA, and of its validation (Introduction: lines 64-75). Furthermore, following the Reviewer's suggestion, we have discussed more extensively the differences of anthropometric variables, the waist circumference in particular (Discussion: lines 26-29).

> BIVA, hand-grip strength, mini-nutritional assessment, and physical capacity assessment have been performed. However, the results were only presented in the table, and the relationship between the TCC and the control group has not been analyzed and discussed.

- Actually, we had already shown the results of BIVA (page 4, lines 161-164; page 7, lines 11-15 of previous version), hand-grip strength (page 7, lines 7-8, 11), MNA (page 4, lines 158-159), and physical capacity assessment (page 7, line 9), and discussed them (BIVA: page 9, lines 4-9, 20-23, 25-26, 39-42; hand-grip strength, page 9, line 3, 36; physical capacity, page 9, line 3). These results had been also considered in the discussion, that was mainly focussed on bioelectrical results (i.e., BIVA). However, considering the Reviewer's comment, the revised discussion has been considerably extended.

> In Table 1, the phase angle TTC of the female was 5.9 degrees, the control group was 6.1 degrees, the Arm phase angle of the male TTC was 5.8 degrees, the control group was 6.8 degrees, the male TTC leg phase angle was 6.1 degrees, and the control group was 6.9 degrees. Some TTCs in the table have better body angle than the control group, and some did not. Most studies have shown that the healthier the body, the higher the cell viability, but the results here need further discussion and clarification.

- We thank the reviewer for this relevant comment and agree with the usefulness of discussing the phase angle with more detail. However, we had noticed that among TCC practitioners muscle mass is lower in the arm (as indicated by phase angle values) and higher in the trunk (line 198 of the previous version). These are the only statistically significant differences in phase angle. As to the total body and the leg, we can only state that TCC and control group show similar characteristics. Considering the Reviewer's concern, we have added a comment on this point in the new version of the manuscript (lines 35-36), highlithing that TCC practice is not associated with muscle mass differences in our sample. Indeed, our results overall show that the differences between TCC and controls mainly refer to fat mass (as indicated by vector length). This result is in line with the literature and has been stressed in the new title.

> Please provide the IRB number.

- The IRB number has been provided in the new version of the manuscript (Methods: line 97).

> BIA test conditions

- As recommended, we have described in more detail the procedure used for BIVA experiments and added a reference to the ESPEN guidelines (Kyle et al., 2004) (Methods: lines 113-114).

> In the BIA test conditions, although it was stated that the experiment was performed in the morning, drinking and diet control, and urination control. When conducting BIVA experiments, there are still many restrictions that need to be clarified, and the author is recommended to enhance the related narrative.

- As suggested, we have added more information on BIA test conditions, including exclusion criteria and health status of the subjects (Methods: test conditions, lines 113-118, 123; exclusion criteria, lines 92-93).

> Please enhance the drawing quality or resolution.

- In agreement with the commonly accepted quality standards, our figures have a resolution of 300 dpi. We can see them with optimal resolution. Maybe this is a problem that arises during the reviewing process. We will take care of this at the moment of publication, if the article will be accepted. Meanwhile, we are sending a pdf copy of the figures for facilitating your revision.

> For the measurement of the bioimpedance parameters of each limb segment, it must be clearly described, and its reliability and repeatability should be explained.

- As suggested, we have added the information on accuracy and reliability of bioelectrical measurements (Methods: lines 118-122).

> What was the basis for determining the sample size in this study?

We have selected all the individuals practicing TCC in Cagliari (Sardinia, Italy) who where in accordance with inclusion criteria and accepted to partecipate at the the study. As declared within the limitations of the study, the sample of TCC practitioners is not very large. However, each subsample includes more than 30 individuals, that is the minimal size indicated by Gay & Diehl for experimental research (Research Methods for Business and Management. New York: Macmillan. 1992). Furthermore, as suggested by Reviewer #1, in the new version of the manuscript we have added the Cohen's statistics in order to evaluate the effect size of the results.

Reviewer 3 Report

This manuscript entitled “Total and regional body composition among Tai Chi Chuan practitioners” intended to analyze total body and regional body composition in TCC middle-aged and elderly practitioners by means of specific bioelectrical impedance vector analysis. The authors bring an interesting study, but there are still some problems that can’t up this review to a publishing level. Some suggestions are listed in the specific comments below.

Specific comments:

“Total and regional body composition among Tai Chi Chuan practitioners”. Considering a better title for this manuscript. Keywords: ageing; Tai Chi; specific bioelectrical impedance vector analysis (BIVA); body composition. In the introduction section, line 39, change ‘are still poorly studied’ to ‘were still poorly studied’. Line 47, remove ‘in fact’, ‘prevent’, line 51, change ‘show’ to ‘showed’. ‘The studies on regional body composition have been based on skinfold thickness distribution only. Specific bioelectrical impedance vector analysis has never been used.’

More references are needed.

Why there is no any body composition related study utilized Specific bioelectrical impedance vector analysis? Please the authors explain it.

In the methods section, adding a title for 2.2-2.6 showing as subtitles.

Line 95-104, please provide a flow chart for the measuring process for a better understanding.

Line 149-153, written as one paragraph.

In the results part, for table 1, revise it as the three-line table style, same as table 2 in this manuscript.

The good picture quality for Figure 1-3 is needed.

In the discussion section, line 13, the literature focused on body composition in TCC suggests a major effect of TCC…

Line 14-15, remove ‘in fact’ and ‘again’.

Author Response

Thank you for the detailed and useful comments, that helped us to improve the quality of the manuscript.

> Considering a better title for this manuscript.

Following your suggestions, we have changed the title in: Lower percentage of fat mass among Tai Chi Chuan practitioners.

> Keywords: ageing; Tai Chi; specific bioelectrical impedance vector analysis (BIVA); body composition.

- Thank you for the suggestion; we have modified the keywords.

> Line 39, change ‘are still poorly studied’ to ‘were still poorly studied’.

- Revised, thank you.

> Line 47, remove ‘in fact’, ‘prevent’

- Revised, thank you.

> Line 51, change ‘show’ to ‘showed’.

- Revised, thank you.

> The studies on regional body composition have been based on skinfold thickness distribution only. [...] More references are needed.

- We have repeated the bibliographic search in order to check if some contribution on regional body composition in TCC was missing, but we have found a new article only (Lan et al., 1998). This study has been included in the discussion of the revised article (Discussion: line 38). We would appreciate if the reviewer could suggest other articles he is aware.

> Why there is no any body composition related study utilized specific bioelectrical impedance vector analysis? Please the authors explain it.

- Specific BIVA is a recently proposed technique and probably not still known by researchers, maybe because slightly difficult to be correctly understood. Indeed, this is the first time it is applied in TCC practitioners. However, the technique has been considered promising, especially in sport sciences (Castizo-Olier et al., 2018). The technique has been already applied to evaluate and monitor body composition changes in several studies (e.g., cavers: Antoni et al., 2017; various athletes: Marini et al., 2019; soccer players: Campa et al., 2020).

Following this suggestion, and as also suggested by Reviewer#1, we have added a more detailed and hopely clearer explanation of specific BIVA and of its validation (Introduction: lines 64-71; 72-75).

> In the methods section, adding a title for 2.2-2.6 showing as subtitles.

- Revised, thank you.

> Line 95-104, please provide a flow chart for the measuring process for a better understanding.

- As suggested, we have added a new figure (fig. 1) with the measurements process flow chart.

> Line 149-153, written as one paragraph.

- We are sorry, but we do not understand this comment. In our version of the manuscript, lines 149-153 include the subtitle 'Statistical analysis'. Does the reviewer mean we should eliminate that title?

>In the results part, for table 1, revise it as the three-line table style, same as table 2 in this manuscript.

- Revised, thank you

>The good picture quality for Figure 1-3 is needed.

- Our figures have a resolution of 300 dpi, that is are in agreement with the commonly accepted quality standards. We can see them with optimal resolution. Maybe this is a problem that arises only during the reviewing process. We will take care of this at the moment of publication, if the article will be accepted. Meanwhile, we are sending a pdf copy of the figures for facilitating your revision.

> In the discussion section, line 13, the literature focused on body composition in TCC suggests a major effect of TCC…

- Amended the refuse, thank you.

> Line 14-15, remove ‘in fact’ and ‘again’.

- Revised, thank you.

Round 2

Reviewer 1 Report

Thank you very much for answering each of the questions I raised.

Reviewer 2 Report

Main Comment:

Regarding the " The body composition items mentioned above were not actually measured in this study. So a considerable amount of correction is needed in your discussion section. [...] we can only see the effect of TCC on anthropometric measurement, waist, arm, calf). It is suggested that the title, abstract, discussion and conclusions of this study need to be significantly revised." in the question I suggested to the author last time, I can understand the author's response to the related question. I understand that BIVA has a certain correlation with body composition. However, whether BIVA can be equivalent or replaced as a direct measurement of body composition requires more direct evidence and methods to confirm. Secondly, the study was renamed "Lower percentage of fat mass among Tai Chi Chuan practitioners". From the results in the manuscript, it can be confirmed that there was a difference between the BIVA of the TCC and the BIVA of the control group. But I still have no way to confirm the scientific evidence that TTC can reduce the fat mass from the information in the author's manuscript. If you really want to verify or the difference between the body composition of the TCC and the control group, then why not use the direct method of DXA.

In the revised manuscript, the author cited many of his own literature to illustrate the relationship between BIVA and body composition. In addition, with the existing literature, I still have high doubts as to whether there was enough evidence to verify or explain the difference in body composition that BIVA can use for each limb. I think, the authors should also agree that if this experiment can increase the verification results of DXA, the research will become quite scientifically meaningful. When planning an experimental study, you need to consider carefully, of course, it also includes planning for the number of samples. It is suggested that the author should make it clear in the manuscript, not just citations or simple statements to indicate consideration.

Minor:

Line 66: "phase angle: arctan Xc / R180 /" I can understand what you want to express, but such expression was obviously problematic. Same problem with Line 144.

Line 120 vs Line 121: "intra-observer technical error of measurement (TEM) and the % TEM" in Line 120, but in Line 121 showed "TEM% = 0.6%", the "%" sign presented inconsistent.

Footnote to Table 1. "BMI: body mass index; Rsp: specific resistance" vs the footnote in Table2. "WAT = workload at anaerobic threshold"; The abbreviated labeling symbols in Table 1, Table 2 were different and need to be unified.

The unit of VO2 in Table 2 was labeled “mL / kg. min-1”, not “mL / kg. min-1”. "VO2" should be instead of "VO2", this error needs to be corrected.

In Line 76. Missing a ")".

Reviewer 3 Report

The authors have made a good revision, it's suitable to be published now.